# The Characterization Pattern of Overburden Deformation with Distributed Optical Fiber Sensing: An Analogue Model Test and Extensional Analysis

**DOI:** 10.3390/s20247215

**Published:** 2020-12-16

**Authors:** Qiang Yuan, Jing Chai, Yiwei Ren, Yongliang Liu

**Affiliations:** 1State Key Laboratory of Coal Mine Disaster Dynamics and Control, Chongqing University, Chongqing 400044, China; renyiwei@cqu.edu.cn; 2College of Energy Engineering, Xi’an University of Science and Technology, Xi’an 710054, China; chaij@xust.edu.cn (J.C.); ylliu.xust@gmail.com (Y.L.)

**Keywords:** distributed optical fiber sensing, characterization, overburden deformation, strain distribution, analogue model test

## Abstract

The evolution of overburden deformation is crucial for safety and environmental efficiency and its monitoring is becoming a key scientific issue. The use of an optical fiber sensor (OFS) for mining engineering is now receiving praise by virtue of its distinct abilities of distribution, high accuracy, and anti-interference measurement. Nevertheless, the dynamic response of OFS monitoring on overburden deformation still needs to be characterized in detail. This paper analyzed the characterization pattern of overburden deformation based on distributed optical fiber sensing (DOFS) by means of an analogue model test. Then, we discuss the influence of rules of optical fiber embedding on a model test in a numerical simulation. The results show that the DOFS monitoring demonstrates the time-space evolution of overburden deformation and the development of three horizontal areas and three vertical zones. A standardization DOFS characterization model is proposed to expound the characterization mechanism of the overburden structure zoning process; the influence of optical fiber embedding on rock displacement in the model test is revealed, and it is found that the displacement error will increase sharply when the fiber diameter is larger than 2 mm. These findings could provide an effective solution for a monitoring method in intelligent mining from the perspective of a theoretical basis and technological system.

## 1. Introduction

Underground coal mine exploitation triggers large deformation and extensive movement of rock mass overlying the coal seam, which accompanies the redistribution of internal stress of overburden stratum rock mass. This kinetic process of mining-induced rock failure is the primary factor that impacts mining safety and ecological protection [1,2]. Therefore, the evolution law of mining overburden deformation has always been one of the most important scientific problems in this field [3,4,5]. In order to solve the difficulties of underground structural monitoring, a series of monitoring methods for mining overburden rock mechanics based on geophysics techniques have been derived [6,7,8]. These have provided solid scientific foundation for studying the movement law of stope surrounding rocks [9], the definition of the water conducting zone [10], and the disaster-causing mechanism of rock dynamic mechanics [11]. However, with the continuous advance of the global industrial intelligent construction tide, the application of traditional means is starting to become powerless, especially in terms of automation and intelligence. It obstructs the innovation development and upgrading of the coal mining industry which needs an advanced method based on an intelligent and intrinsic safety system as a new impetus.

Since the widespread application of distributed optical fiber sensing (DOFS) technology into geotechnical and geology engineering [12,13], its abilities for distribution, high accuracy and anti-interference measurement have widely and deeply improved the engineering monitoring efficiency [14,15,16]. By virtue of these unique advantages of realizing real-time online monitoring and intelligent remote control [17,18], it becomes an optional solution to the intelligent mining construction and has tremendous potential for application in mining engineering [19,20]. The preliminary application of DOFS in mining rock mechanics and ground control research [21,22] already indicated that it realized the accurate capture and distribution monitoring of internal rock stress [23,24]. Moreover, the DOFS technology has the advantages of portable installation, low environmental requirements, large-scale, and dynamic continuous measurement, which are specific to the deformation characteristics of mining-induced overburden study. More targeted research has been launched. Zhao C. et al. [25] proposed a method to monitor the coal mine roadway roof separation by fiber Bragg gratings (FBG) displacement sensors and studied the variation trends of roof separation displacement. Zhao Y. et al. [26] developed an FBG monitoring system for overburden control in underground coal mining to accurately capture the dynamic impacts of progressive face advance on roof displacement and rock mass stress. Nan S. et al. [27] investigated the stability of mine stope during the whole mining process by Brillouin optical time domain refectory (BOTDR). Piao C. et al. [28,29] studied the deformation features of the overburden layer around a vertical mine shaft by FBG and Brillouin optical time domain analysis (BOTDA) technologies, then investigated the strain distribution rule and movement characteristics of the stratum under reamer-pillar mining. Zhang D. et al. [30,31] revealed the relationship between the strain distribution and the overburden stratum deformation based on BOTDR monitoring via direct drilling of optical cables implantation, and more importantly it delineated the water conducting zone. Sun B. et al. [32] performed dynamic detection and analysis of overburden deformation based on BOTDR to capture the zoning evolution of the caving zone and fractured zone. Zhang P. et al. [33] discussed the law of overburden deformation in the section space of rocks surrounding a roadway by laying a BOTDR-based distribution fiber sensor in the drill hole. Hou G. et al. [34] proved the feasibility of optical fibers for monitoring overburden movement and obtained the law of strata fracturing and collapse based on BOTDR technology. Chai J. et al. [35,36] analyzed the basic law of Brillouin frequency shifts compared to overburden destruction form to study the overburden deformation characteristics based on BOTDA technology. The zoning features and key stratum broken rules were discussed. Liu Q. et al. [37] studied the deformation tendency and the supporting effect of the crossed rock pillar to the surrounding rock by analyzing the continuous monitoring on surrounding rock based on optical frequency domain reflectometry (OFDR)-based sensing technology. Liu J. et al. [38] carried out both laboratory experiment and field application to study the soil deformation response to water draining-recharging conditions based on Brillouin optical frequency domain analysis (BOFDA) technology in long-term monitoring.

Consequently, the advantages of DOFS monitoring in mining-induced overburden rock deformation research have gradually unfolded. It is demonstrable that relevant research is being diverted to study the characterization mechanism of DOFS monitoring rather than staying in the realm of applicability validation. Researchers are beginning to report the comparison relations of the overburden deformation and the distributed strain characteristics, such as the horizontal three zones formation of overburden deformation had been detected by FBG sensors [39]. The variation in time and space of the strain distribution of overburden deformation and the height of fractured zone were captured by BOTDR [40]. The conceptual model for describing the overburden deformation process with BOTDA features which explicitly identified the zoning development of overburden was established [41]. The stratum roof weighting and rock structure forming evolution were demonstrated based on BOFDA and close-range photography processing [42]. Because of this progress, it is necessary to figure out the fundamental mechanism of the dynamic response characteristics of DOFS in overburden deformation stimulation. For that purpose, this paper offers further insight into the DOFS characterization mechanism of the mining-induced overburden deformation by contrasting the analogue model test and numerical simulation based on previous studies.

## 2. Analogue Model Test of Overburden Deformation by DOFS Monitoring

### 2.1. Basic Principle of BOTDA Technology

The BOTDA is a superior technology of the DOFS system, which was put forward by Horiguchi in 1989 [43]. It is based on a theoretical method of applying the Brillouin scattering principle, the system structure and sensing principle of which are shown in Figure 1.

By using two light sources as the pumping pulse light and the detecting continuous light, in which the detection of signals can be Brillouin gain or loss of Brillouin signal. When part of the optical fiber strain or temperature changes, the corresponding Brillouin frequency shift (BFS) changes accordingly. The Tuning makes the frequency offset of two pumping light equal to the new BFS, so that it will be able to receive the Brillouin scattering signal. By detecting the power of continuous light out of the optical fiber, the corresponding frequency offset can be determined on the fiber when its energy transfer is maximized maximum in every small area. As a result, the information about temperature and strain distribution can be determined, which realizes distributed sensing. Moreover, the distribution of temperature and strain could be obtained through the signal power of the normalized change value along the optical fiber.

At present, the BOTDA technology already has a relatively high testing accuracy and spatial resolution. To be specific, by means of a DOFS demodulator of NBX-6055, the measurement accuracy and spatial resolution can reach ±7 με and 5 cm, and it will fulfil the requirement of a similar material model test by setting a sampling interval of 1 cm. Even though measurement of both ends of BOTDA demands more attention to optical fiber protection, the good survival environment of an indoor experiment compared to a field test means it is becoming the preferred DOFS system for experimental study.

### 2.2. Setup of Analogue Model and DOFS Monitoring System

The rock mass failure in an underground mining area cannot be directly captured due to the massive movement of overburden usually being beyond the working space, which means that the study of mining engineering is a black box issue. A similar material model test is an analogous model of mining rocks to solve the overburden deformation or rock structural evolution problems. It had turned into one of the most effective solutions in mining engineering since it makes the rock stratum visible.

In summary, an analogue model of overburden stratum constructed by similar material was built in a laboratory. It has a length of 300 cm, height of 128 cm, and width of 20 cm. The height of the floor rock stratum is 5 cm, the height of the coal seam is 5 cm, and the height of the overburden stratum is 118 cm. As shown in Table 1, the similar material was comprised of sand, gypsum and mica powder. Then they were mixed with certain material ratios which were calculated according to similarity theory. The simulation of coal seam excavation started at the left side of the model and two coal pillars of 30 cm length were set to eliminate the boundary effect on both sides. Therefore, there are 80 excavation steps with a distance of 3 cm between them. The detail experimental setup is shown in Figure 2. The monitoring system included the NBX-6055 optical stress analyzer, the Φ2 mm normal single-mode optical fiber sensors, the FBG (Fiber Bragg Gratings) sensors and the dial gages. The optical fiber is in the polyurethane cable of the tight buffer fibers wrapped within an elastic protective jacket. The optical fiber is installed by fixing it on the model frame before the model was built. Its upper end was suspended and stretched in a vertical state by pre-stress at 25 N. Thus, they were vertical in the model and went through the whole overburden rock stratum, and this is also a simulation of the geophysics instrument installed through surface drilling in the field test.

Three optical fibers F1, F2, and F3 were installed. The cable length was 118 cm, which is equal to the height of overburden stratum and set up with three FBG sensors and three dial gages that are mounted near F2. The measurement of FBG sensor and dial gage was used for accuracy testing compared to DOFS measurement, and the related results [43] showed that the BOTDA measurement precisely captured the rock mass deformation.

## 3. Measurement Results Analysis

### 3.1. Strain Profiles

Figure 3 shows the strain profiles of DOFS monitoring. The ordinate is the thickness of overburden stratum, the abscissa is the measured strain, in which the positive strain represents the tensile stress and the negative strain represents the compressive stress.

It can be found that the strain distribution is obviously affected by mining influence; the strain profile changes regularly with the mining distance. The working face was mined from left to right so that the optical fiber F1 will firstly react with mining influence, then F2 and F3. Meanwhile, the development and variation tendency of each strain profile could be summarized as follows. The monitored strain will basically keep its original state when there is no mining influence, which means the working face is far away from the optical fiber. Because the working face advances closely to the rock mass where the optical fiber is located, the strain profile changes negatively with small amplitude; however, when the working face advances to the very place where the optical fiber is located, the strain profile shows a large positive change. Eventually, the strain profile will be in a stable negative form when the working face advances far away again from the optical fiber.

Besides the distribution law of the strain profile mentioned above, it also can be seen that the strain profile shows regular changes with the height of overburden. In particular, when the working face is advanced around the optical fiber, there will emerge a step-form changes in the strain profile and this will expand with the increasing of the height of the overburden stratum. It follows that the strain profile will change differently when the mining influence affects the rock mass under different locations of working face. Meanwhile, the different height ranges of the mining overburden stratum deformation in the same mining distance will lead to strain profile changes on different height of the overburden stratum, which means the DOFS monitoring has acquired the space and time distribution characteristics of overburden deformation. The strain profile of optical fiber can be used to invert the deformation and failure law of mining the overburden stratum.

### 3.2. Disassemble Analysis of Strain Distribution

In order to reveal the strain distribution characteristics of different mining periods to invert the overburden deformation, the strain profile was disassembled based on the change of mining distance into further analysis. Taking the measurement results of optical fiber F1 as an example, the strain profile and mining overburden deformation states are compared to analysis under the different distance of mining distances and the optical fiber locations, which is shown in Figure 4.

When the mining distance is 0–36 cm, the distance between the working face and optical fiber location is 60–24 cm. As an additional remark, this means that the working face advances close to the optical fiber if the distance is positive and the working face advances away from the optical fiber if the distance is negative. It can be seen that the strain distribution is basically around zero, which indicates that the overburden stratum monitored by the optical fiber has not been affected by mining activity, and the rock mass stress does not change at all; thus, it remains in the primary rock stress area.

When the mining distance is 39–57 cm, the distance between the working face and optical fiber location is 21–3 cm, the working face advances close to the optical fiber. The characteristic of the strain profile in this type B is that the strain distribution is stable to negative (about −1000 με) over a range of overburden height (0–25 cm), and part of the strain decreases with the increasing of overburden height. That is to say that the monitored overburden stratum is affected by mining influence and there exists some compressive stress in the rock mass. Combined with the theory of three horizontal zones in mining-induced rock mechanics, there exists an area of leading abutment pressure in front of the working face. The stress distribution characteristic is the rock mass that will produce concentrated compressive stress within a certain overburden height range, and this will decrease with the increasing of the overburden height. As a result, the strain distribution simply verifies the stress characteristics of rock mass in the leading abutment pressure area.

When the mining distance is 60 to 117 cm, the distance between the working face and optical fiber location is 0−57 cm. The characteristic of the strain profile in type C is that the strain distribution presents a step-form development; the strain value is basically positive, and it remains constant over a range of overburden height. Further investigation shows that the strain remains unchanged in the height range of 0–20 cm, but the height range of the strain distribution increases with increasing mining distance above the height of 20 cm. Thus, it can be seen that the strain distribution curve changes in at least three steps. Similarly, combined with the theory of three horizontal zones, it reveals that the overburden stratum is in the fracture and separation area at this time, its range will develop and expand upward due to the rock mass becoming gradually fractured, and break downward into the gob under the mining influence. Therefore, the strain distribution in type C represents the stress characteristics of rock mass in the fracture and separation area. Also, it is worth noting that the overburden deformation in that area will generate three vertical zones, and the internal stress characteristics in those different zones have typical characteristics that distinguish them.

When the mining distance is 120 to 240 cm, the distance between the working face and optical fiber location is −60 to −180 cm. The characteristic of the strain profile in this type D is that the strain value is all negative, and the strain distribution will gradually increase at first and then eventually remain constant as the working face advances, which means the monitoring rock mass is conducted by compressive stress and it will increase until it is stable. It is obvious to see that the strain distribution in type D corresponds to the stress characteristics of the rock mass in the recompacting area, the mining influence will remain steady to form the mining stability zone due to the compaction filling of overburden stratum under gravity.

The above analysis disassembled the whole mining process into four parts, such that the strain distribution characteristics of each part are triggered by the specific overburden deformation process, thus corresponding to four different types of overburden deformation characteristics. Meanwhile, these parts also describe the performance of the three horizontal overburden zones in mining rock mechanics. In conclusion, the development of the strain distribution is closely related to the mining overburden deformation, and the strain distribution explains the formation and evolution mechanism of the three horizontal overburden zones from the perspective of time.

### 3.3. Analysis of Strain Distribution Characteristics

According to the analysis of strain distribution characteristics in type C of overburden deformation, the rock stratum in the fracture and separation areas will break upwards to a certain range of overburden height, then the strain profile will change at different height ranges. As a consequence, these strain profiles in the fracture and separation area are demonstrated to analyze the development law of strain distribution characteristics.

As shown in Figure 5, it can be seen that the strain profile in the type C period demonstrated obvious step-form evolution characteristics, which are composed of three characteristic regions and two interface lines. To be specific, the characteristic region 1 of the strain profile shows that the strain value gradually increases within a small range or remains constant within a certain range, and each strain distribution height range also remains constant. Thus, the first step in the strain curve is formed, and it has a stable shape and does not change with the mining distances.

The strain distribution will increase significantly over a certain height range in the characteristic region 2. The strain value also increases as the mining distance increases. Meanwhile, the height range of the strain distribution develops, becoming higher, to form the second step, and it will keep going up on the overburden height with the advancing of the working face. Furthermore, strain value in characteristic region 2 will significantly increase compared to the characteristics of region 1, so that there will be an interface line 1 which distinguishes between characteristic regions 1 and 2.

The strain profile of characteristic region 3 shows that the strain distribution in the overburden height is attenuated, that is, the strain value decreases with the increase of overburden height. At the same time, the strain value in characteristic region 3 suddenly decreases compared to that in characteristic region 2, thus forming the interface line 2, which is the boundary between feature regions 2 and 3.

The gradual deformation and movement of the overburden stratum in the fracture and separation area will develop upward towards the overburden height. Thus, three vertical zones will be produced in the overburden stratum, namely the caving zone, fissure zone and bending zone. The stress characteristics of the rock mass in those three zones can be expounded as follows: the rock mass is heavily damaged in the caving zone on the impact of mining-induced stress, then the overburden stratum will break into blocks and move downward to the mining gob. The internal stress of rock mass at this time is in a transformation state, within which the stress will be conducting from tensile stress to compressive stress. The tensile stress of the rock mass in the caving zone decreases gradually when the overburden stratum is divided into a fracture and separation area, and it will transform to compressive stress when the overburden stratum belongs to recompacting area. A large number of fractures and fissures develop in the rock mass of the overburden fissure zone, the entire overburden stratum will be under great tensile stress on the impact of mining-induced stress. The overburden stratum of the bending zone has a bending deformation under the action of mining-induced stress, and its bending degree decreases with increasing overburden height, so that the tensile stress of the rock mass also decreases towards the overburden height.

Overall, it is evident that the characteristic regions are typical of three vertical zones of mining overburden stratum. The deformation and structural evolution of the overburden stratum leads to zoning performance that results in the step-form of strain distribution, so that the step-form will display three steps and the height of each step will correspond to the individual height of each vertical zone of overburden deformation. To demonstrate the characteristics above, the height of the mining three vertical overburden zones and the height of the three steps in the strain profiles are demonstrated comparatively in Figure 6. The strain distribution curves of optical fiber F1 and F2 are taken as examples to analyze since the step form of F3 is underdeveloped because the boundary conditions affect the overburden deformation. In Figure 6, D represents the direct measurement value of the three vertical zones’ height, S is the height of the three-step strain profiles.

It can clearly be found that the both the height of the vertical three zones and the three-step form are consistent with each other, which proved that the three-step characteristics of the strain profile are the stress and strain characteristics of overburden rock mass in different mining influence areas, also the characteristics of monitored rock mass in three different vertical overburden zones. In conclusion, the DOFS monitoring results reflected the operation of mining influence in the overburden space, and the strain distribution expounded the formation and evolution mechanism of the three vertical overburden zones from the spatial perspective.

## 4. Discussion of DOFS Monitoring for Overburden Deformation

### 4.1. Comparison of Supplement Analogue Model Tests

#### 4.1.1. Experimental Setup

It is evident that the DOFS realized the monitoring of overburden deformation and its structure evolution process, so that an additional two model tests were carried out in order to add credence to the monitoring mechanism that is analyzed above. The same DOFS monitoring scheme was implemented in the supplementary experiment.

Figure 7 shows the analogous models and the DOFS system setup. The overburden height of both models is 94 cm and 115 cm, the simulation excavation starts from right to left with the mining interval of 2 cm. We mounted two optical fibers of A-F1 and A-F2 in model A and three optical fibers of B-F1, B-F2 and B-F2′ in model B. The B-F2 and B-F2′ are installed in close range such that the distance between them is less than 5 cm.

The purpose of that arrangement is that the previous analysis can be verified by monitoring results of A-F1 and B-F1. The result of close range installed optical fibers B-F2 and B-F2′ can demonstrate the DOFS characterization of overburden structure evolution in same location by two optical fibers and verify the repeatability.

#### 4.1.2. Verification of Overburden Deformation Development

Figure 8 shows the strain profile of two supplementary model tests. It can be seen that the strain profiles of optical fiber A-F1 and B-F1 also display the same monitoring results of the primary model test, which can be divided into four parts and corresponded to four different types of overburden deformation forms.

The strain distribution can be also decomposed into four parts that represent the overburden deformation characteristics of the monitored rock mass in an area not influenced by mining, a leading abutment pressure area, a fracture and separation area, and a recompacting area. In other words, it is the characterization of overburden deformation in three horizontal zones. Figure 9 presents the strain distribution when the overburden rock mass is located in the fracture and separation area. There also emerges a three-step formation which can be decomposed into three parts that represent the overburden deformation characteristics of the monitored rock mass in a caving zone, fractured zone, and bending zone. This is the characterization of overburden deformation in three vertical zones.

As a result, it follows that the strain distribution obtained by DOFS indeed represents the time-space evolution of mining-induced overburden deformation, the characteristics of strain distribution demonstrated the whole process of stratum rock deformation and detailed features of stratum structure development.

#### 4.1.3. Reliability Validation of DOFS Monitoring

The reliability of DOFS measurement is validated by comparing two optical fibers in almost the same location of the overburden stratum, which are the B-F2 and B-F2′. Given that the mining influence will affect the overburden stratum around optical fiber B-F2′ when the working face approaches B-F2, if the strain profiles have a high consistency and tiny dissimilarity, the reliability could be affirmative.

Figure 10 shows the comparison results of strain profiles, which indicate that the strain distribution has the same developmental tendency. It also can be found that the development of the strain profile of B-F2′ lags behind that of B-F2 at the first mining period because the distance between optical fiber B-F2′ and the working face is larger than the distance between B-F2 and the working face. However, they will eventually evolve in the same distribution. Specifically, the strain profile shows the same negative value of strain when the mining distance is 94 cm, which means the surrounding rock mass gets into the overburden deformation of the leading abutment pressure area. The step-form of the strain distribution is also displayed when the mining distance is 112 and 130 cm, but such step-form is not completely developed at the mining distance of 112 cm and will be fully illustrated till the mining distance arrives at 130 cm. Finally, when the surrounding rock mass of the overburden stratum is in the recompacting area, i.e., the location of the mining stability zone of overburden stratum with the optical fibers, then the same strain distribution of two optical fibers will be produced.

Based on these results, it is found that the strain distribution corresponds exactly to the mining influence distribution caused by the distance error effect of two adjacent embedded optical fibers. Therefore, it proves that the measurement of two optical fibers will develop the same profile, which suggests the DOFS measurement has a superior reliability.

### 4.2. Standardization Model of Overburden Deformation Based on DOFS Monitoring

According to the preceding analysis, a standardization characterization model of overburden deformation based on strain distribution development is proposed in Figure 11. The red line is the strain distribution of DOFS monitoring under each specific circumstance of overburden deformation forms. These forms are displayed on the left, and they are divided into three parts in terms of the characteristics of the three horizontal mining overburden areas.

Figure 11a shows the overburden deformation characteristics when the overburden rock mass is located in the leading abutment pressure area and its corresponding strain distribution is based on DOFS monitoring. The overburden deformation can be summarized as the abutment pressure effect section and no mining influence section, which is consistent with the characteristics of overburden deformation of type B. To be specific, the strain distribution will result in negative strain in a certain range of overburden due to the concentration stress (i.e., macro performed as abutment pressure) development in the section a-i, and the strain value will decrease with the height. In section a-ii, the strain distribution will retain its original values due to the mining influence return to the original status as overburden height increases.

Figure 11b shows the overburden deformation characteristics when the overburden rock mass is located in the fracture and separation area and its corresponding strain distribution based on DOFS monitoring. The overburden deformation can be summarized as caving and collapse section, fracture and separation section, bending movement section, and no mining influence section, which is consistent with the characteristics of overburden deformation of type C, and it can be concluded as the three vertical overburden zones characteristics. Detailed analysis is as follows: the section b-i indicates the overburden deformation of the caving zone and its strain distribution, the strain range is in keeping with the height of the caving zone and the strain remains at a stable positive value. Section b-ii indicates the overburden deformation of the fractured zone and its strain distribution, the strain distribution corresponds to the overburden deformation but the strain value is evidently larger than that in the caving zone. Section b-iii indicates the overburden deformation of the bending zone and its strain distribution, the strain value remains positive and it will decrease as overburden height increases. Section b-iv indicates the overburden deformation in an area not influenced by mining, and the strain distribution will stay in its original status like in section a-ii.

Figure 11c shows the overburden deformation characteristics while the overburden rock mass is located in the recompacting and mining stable area and its corresponding strain distribution is based on DOFS monitoring. The overburden deformation can be summarized as rock mass compressed into an integral section again, and mining influence degrading to a neutral section. This is consistent with the characteristics of overburden deformation of type D. Specifically, expression shows that the overburden rock mass in the caving zone and in part of fractured zone will be compressed under the stress of overlying stratum weighting. It will be recompacted into a stable integral status, which results in the strain being displayed as a stable negative distribution.

This standardization model completely dissects the time-space evolution of the strain distribution in the whole overburden deformation process. This expounds the DOFS characterization mechanism of the mining overburden structure zoning characteristics from the perspective of the three horizontal and vertical zones of deformation of the overburden stratum. Furthermore, other existing results [44] conformed to the standardization model, suggesting that it is suitable to be utilized to demonstrate the mining overburden deformation and structural evolution. In addition, it needs to be noted that the characterization model provided a preliminary definition of the interfaces between these three vertical zones, and more study on the precise definition of the interface location to clearly distinguish between those zones will be done in the future.

### 4.3. Influence of Optical Fiber to Overburden Deformation by Numerical Simulation

Previous studies [45,46] have shown that optical fiber embedding will affect the integral physical and mechanical properties of rock or soil material. As for the analogue model test, geometrical parameters of the optical fiber are probably important specifications that influence the mechanic performance of the analogous model material. Thus, a numerical simulation was conducted to analyze the influence of rules of overburden deformation in an analogue model test by optical fibers with different diameters.

A numerical model of overburden stratum based on ANSYS was constructed based on Table 2 and the finite element model size was set as 900 cm (length) × 20 cm (width) × 200 cm (height) to eliminate the boundary effect and maximally display the deformation status of the overburden rock mass, which is shown in Figure 12. The optical fiber was simulated by the BEAM 188 unit of ANSYS software, which is a three-dimensional linear unit based on the Timoshenko beam theory and can usually be used for simulating the slender beam because it considers the shear deformation and has six degree of freedom in each node. In this case, given that the optical fiber has a much larger length than the cross section, the BEAM 188 is only suitable for simulation. The Drucker-Prager model has been used for defining the overburden stratum material because it is able to simulate the elastoplastic characteristics of rock and soil. Then, a free boundary had been applied to the top side of the model and the other five sides were applied with displacement constraint. Thereafter, the optical fiber was bonded with the overburden stratum and the kill element has been applied to simulate the coal seam excavation.

The Figure 13 shows the displacement analysis under four different optical fiber embedding conditions. It can be concluded that the displacement of overburden stratum with no optical fiber embedded changes the largest, and the displacement will decrease with the increasing of diameter of the optical fiber. The results indicate that the optical fiber embedding will prevent the change of displacement. Moreover, the larger the diameter of optical fiber embedding, the larger the prevention. In another words, the optical fiber affects the natural deformation of overburden in the analogue model test. However, according to the four curves in each figure, it is found that the prevention displacement is less than 5 mm both when the optical fiber is embedding or if there is no optical fiber.

Furthermore, Figure 14 discussed the prevention displacements indicated by displacement error rate under optical fiber embedding condition and no optical fiber condition. The displacement error rate means the rate of different displacements under different optical fiber embedding conditions, it can be calculated by using the displacement under certain optical fiber embedding conditions minus the displacement under no optical fiber embedding conditions. This analysis shows that the error rate of displacement change is less than 5% under conditions of optical fiber embedding with a diameter of 1 mm and 2 mm, but it increases to 12% when the diameter of the embedded optical fiber is 3 mm. As a result, although the optical fiber embedding will inevitably impact the analogous model, it also indicates that the influence can be ignored when the embedded optical fiber diameter is under 2 mm.

Four numerical models were set up in the experiment. The first model embedded no optical fiber, the second model vertically embedded three optical fibers with a diameter of 1 mm, and each three optical fibers with diameters of 2 mm and 3 mm were inserted in the third and fourth model. That is, four different optical fiber embedding conditions are set up for analysis. Each 300 cm length of boundary coal pillars was set due to the boundary effect, so that the total excavation length remains at 300 cm and it will be caved in three steps with each step length being 100 cm. More importantly, six displacement measuring points were marked on the model to discuss the deformation rules.

Figure 15 shows the results of the linear fitting of mining steps and displacements of six measuring points in the overburden stratum, from which it can be seen that the displacement and mining step have a good linear relationship, and the fit coefficient is up to 0.9. However, it can also be seen from the slope of the fitting curve that the degree of overburden deformation under the same excavation step decreases with the increase of optical fiber diameter.

On the basis of the above analysis, the coefficient of primary term (i.e., the slope of fitting curve) of the linear fitting equation is taken and named as the change rate of displacement (CRD) to discuss the influence of the overburden movement and the diameter of the optical fiber. The CRD represents the displacement change value of the overburden deformation incurred by the unit mining step. The closer the CRD of the overburden stratum with the optical fiber is to that without the optical fiber, the less the impact of optical fiber on the overburden deformation is. Furthermore, Figure 16 shows the results when the CRD and the diameter of the optical fiber are brought into the fitting analysis. This demonstrates that the CRD and the diameter are situated in the quadratic fitting relationship, and the influence of optical fiber installation will be continuous on the overburden deformation in the model test experiment. Nevertheless, the relationship between optical fiber diameter and CRD has provided the theoretical basis for optical fiber selection in the experimental studies and may be a reliable technical support for engineering applications of DOFS monitoring.

## 5. Conclusions

This paper studied the characterization pattern of time-space evolution of overburden deformation based on DOFS monitoring by analogue model test and numerical simulation. This not only aims to promote the extended application of the DOFS monitoring method from the perspective of experimental science, but also to provide an effective solution for intelligent mining from the perspective of a theoretical and technological basis. Detailed conclusions are as follows. 

(1) The whole process and full scope dynamic monitoring of the strain distribution of mining-induced overburden deformation is achieved by DOFS monitoring. The strain profiles are dissembled into four parts to explain the development of three horizontal overburden areas in time, and the characteristics of strain distribution are divided into three parts to reveal the evolution of three vertical overburden zones in space.

(2) A standardization characterization model of overburden deformation based on DOFS monitoring is proposed by means of the response mechanism analyzing the strain distribution on mining overburden deformation, which can be dissected into three standard characterization patterns, and fully describes the DOFS characterization of different overburden deformation development under different mining influences based on strain distribution.

(3) It is inevitable that the installation of the optical fiber in experimental application of DOFS monitoring will impact the veracity of analogous model. However, the numerical simulation indicates that the influence can be ignored when the diameter of the embedded optical fiber is less than 2 mm based on the overburden displacement error analysis. Moreover, the CRD is proposed to discuss the relationship of overburden deformation under the condition of different diameters of optical fiber embedding and the condition of no optical fiber embedding, which could be an optical fiber selection guide in the experimental application of DOFS monitoring.

## Figures and Tables

**Figure 1 sensors-20-07215-f001:**
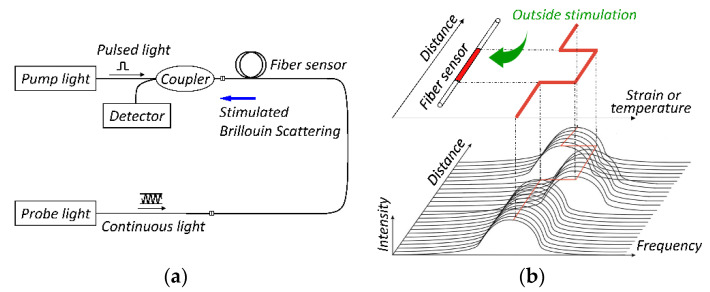
Schematic diagram of Brillouin optical time domain analysis (BOTDA) technology: (**a**) system architecture of both ends detection; (**b**) sensing principle of Brillouin optical time domain analysis.

**Figure 2 sensors-20-07215-f002:**
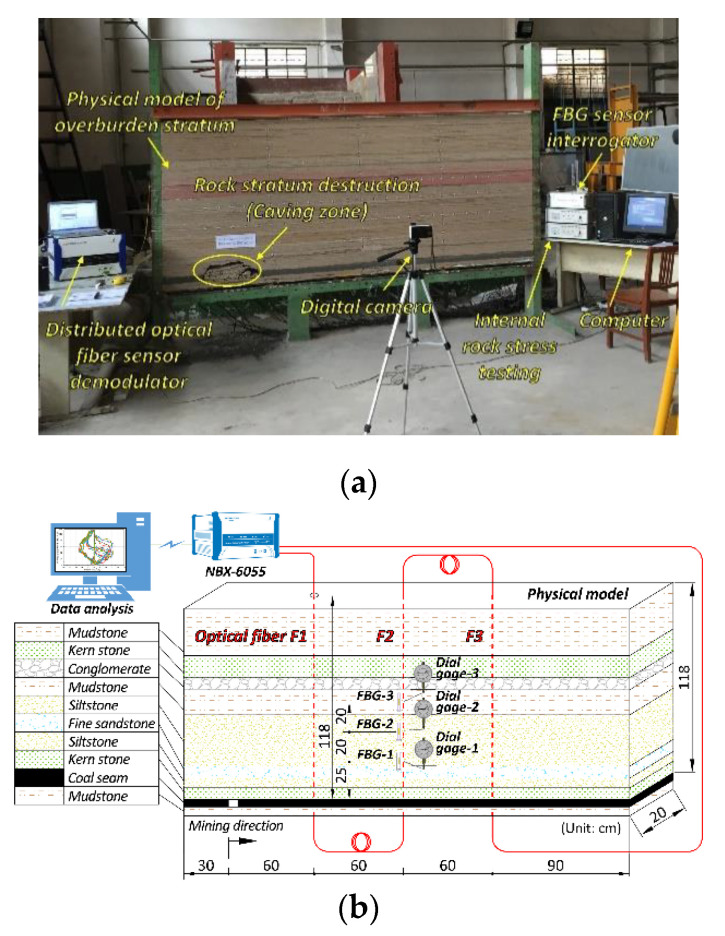
Experiment setup: (**a**) is the analogue model of overburden stratum and (**b**) is the layout of the distributed optical fiber sensing (DOFS) monitoring system within the model test.

**Figure 3 sensors-20-07215-f003:**
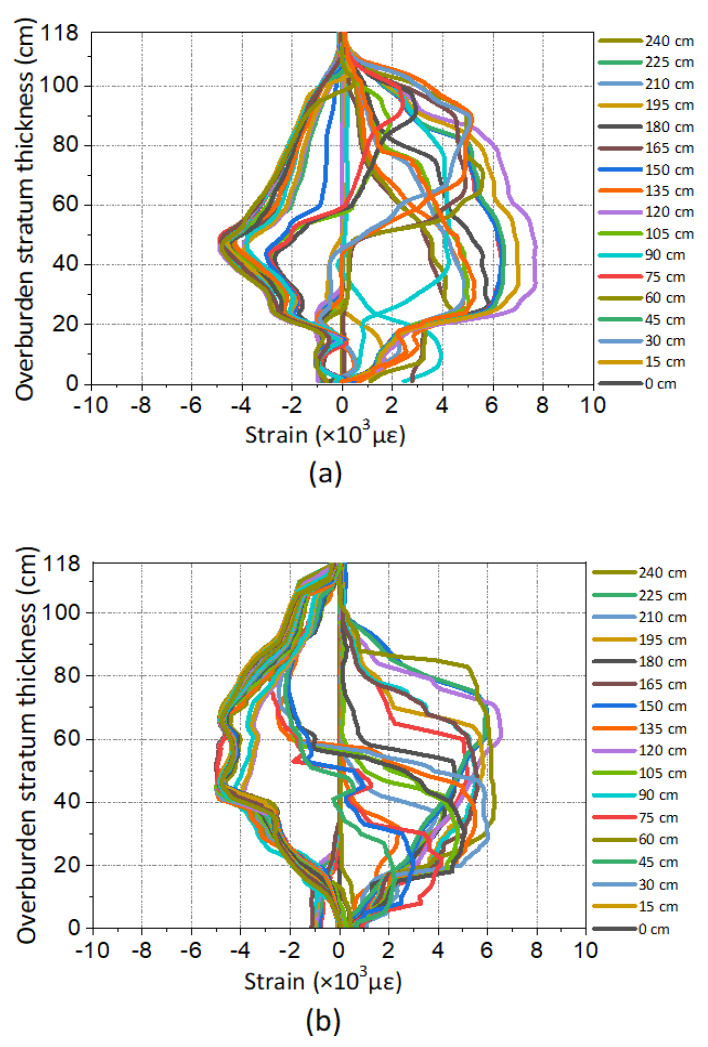
Strain profiles of DOFS monitoring in the model test. The different colored lines refer to the periods of different mining distance: (**a**) is the results of optical fiber F1, (**b**) is the results of optical fiber F2 and (**c**) is the results of optical fiber F3.

**Figure 4 sensors-20-07215-f004:**
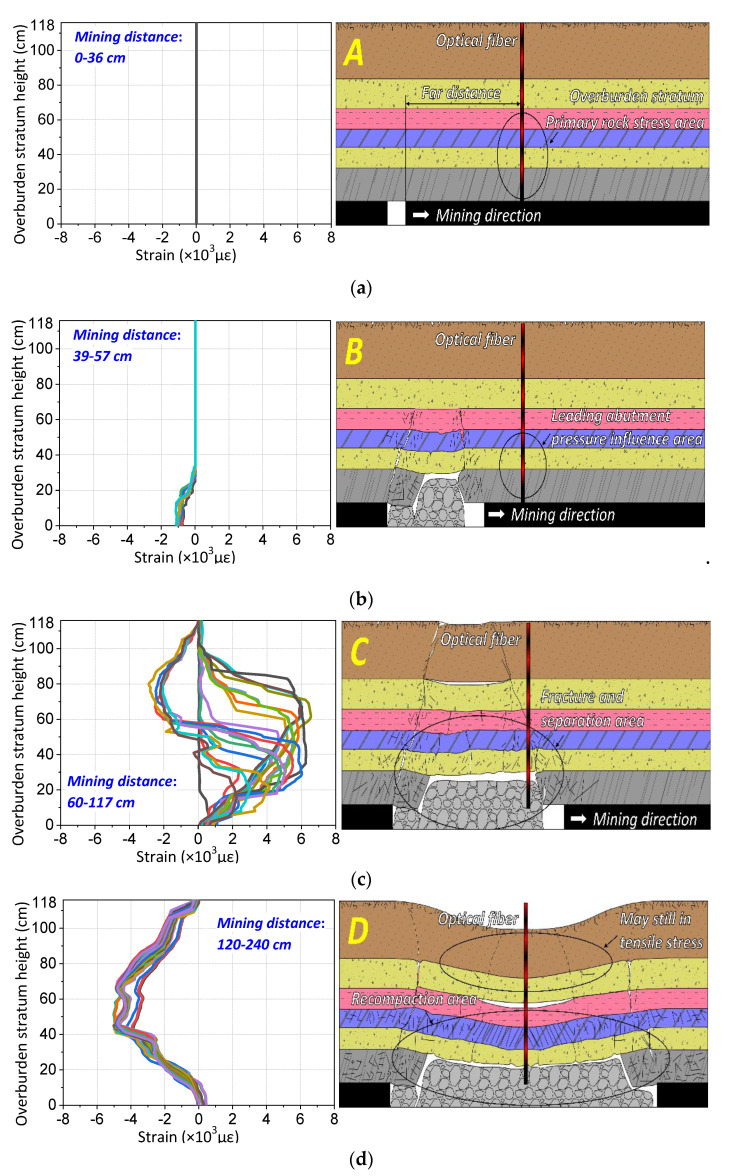
Disassembled strain distribution of F1 versus mining overburden deformation forms. There are four sections of strain distribution and four corresponding deformation forms, (**a**) is the strain distribution compared to the deformation form of type A primary rock stress area, (**b**) is the strain distribution compared to the deformation form of type B leading to the abutment pressure influence area, (**c**) is the strain distribution compared to the deformation form of type C fracture and separation area, and (**d**) is the strain distribution compared to the deformation form of type D recompacting area.

**Figure 5 sensors-20-07215-f005:**
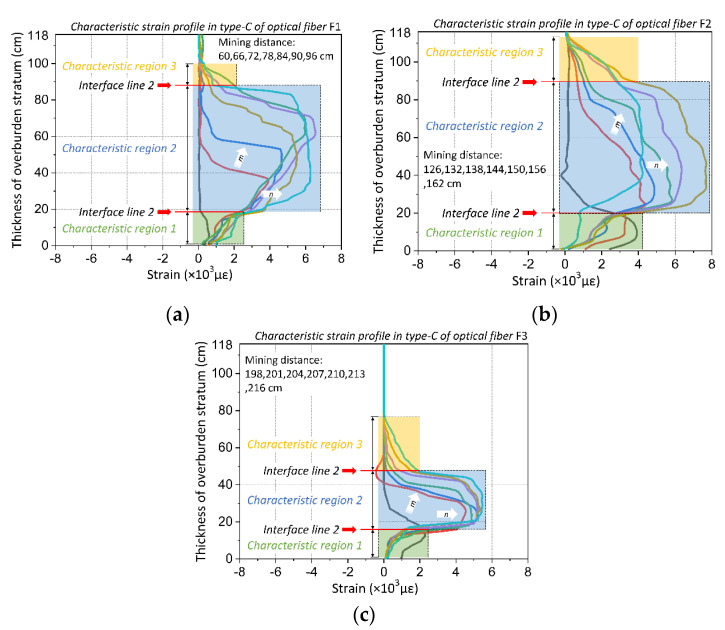
Strain distribution characteristics when the optical fiber located in the fracture and separation area of overburden deformation of type C: (**a**) is the monitoring results of optical fiber F1; (**b**) is the monitoring results of optical fiber F2 and (**c**) is the monitoring results of optical fiber F3.

**Figure 6 sensors-20-07215-f006:**
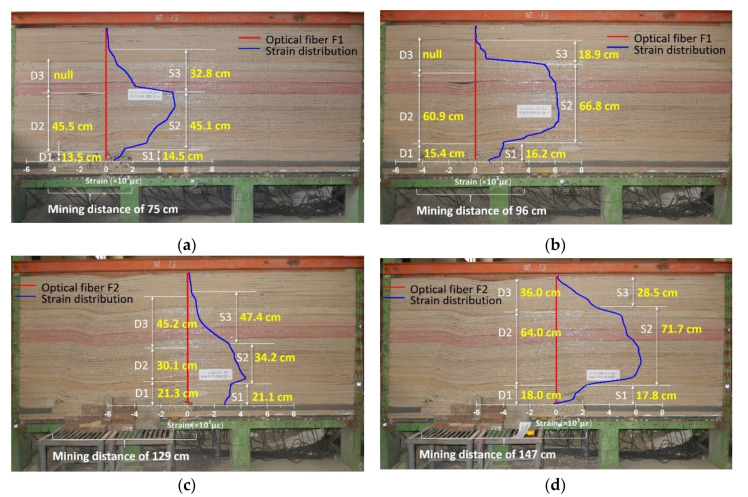
Comparison of mining overburden vertical three zones and strain distribution: (**a**,**b**) are the analysis of overburden deformation structure and strain distribution of optical fiber F1 at the mining distance of 75 and 96 cm; (**c**,**d**) are the analysis of overburden deformation structure and strain distribution of optical fiber F2 at the mining distance of 129 and 147 cm.

**Figure 7 sensors-20-07215-f007:**
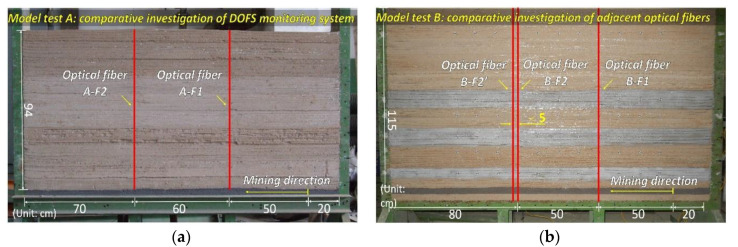
Layout of DOFS system in the comparison experiment of supplement analogue model tests: (**a**) is the setup of supplement model test A; and (**b**) is the setup of supplement model test B.

**Figure 8 sensors-20-07215-f008:**
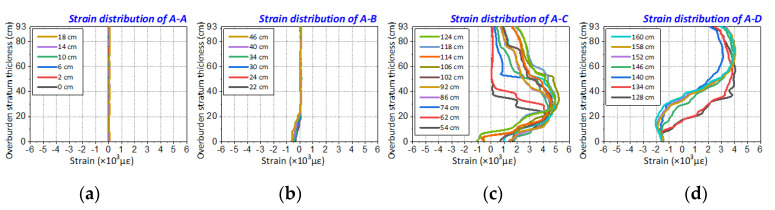
Disassemble analysis of the strain profile in supplementary model tests: the strain distribution of two supplement model tests is divided into four parts, each of those represents four types of overburden deformation forms. (**a**) is the strain distribution of overburden deformation type A in the model test A; (**b**–**d**) are type B, C and D in model test A; Under the same logic analysis, (**e**–**h**) are the strain distribution of type A, B, C and D in model test B.

**Figure 9 sensors-20-07215-f009:**
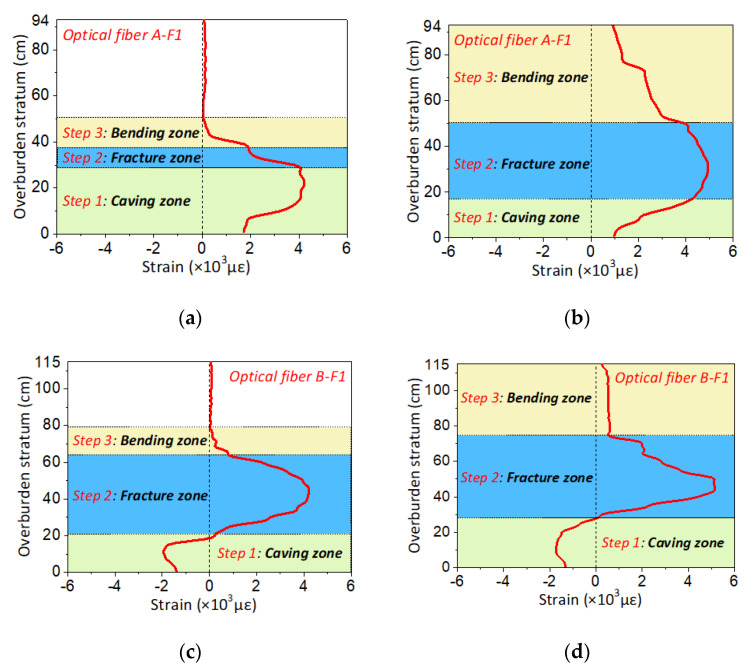
Verification of strain distribution characteristics of step-form: (**a**,**b**) are the characterization of three-step form when the mining distances are 62 and 96 cm based on optical fiber F1 monitoring in model test A; (**c**,**d**) are the characterization of three-step form when the mining distance is 70 and 76 cm based on optical fiber F1 monitoring in model test B.

**Figure 10 sensors-20-07215-f010:**
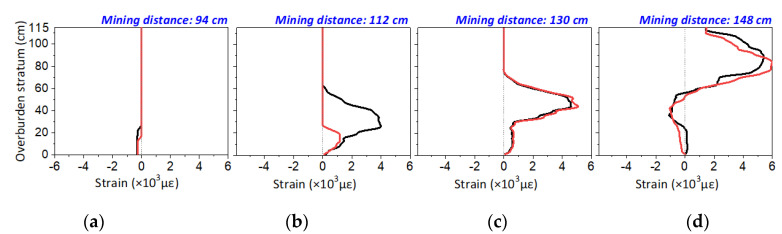
Strain profile comparison of two adjacent embedded optical fibers: the red line is the strain distribution of optical fiber B-F2′ and the black line belongs to optical fiber B-F2. (**a**–**d**) are the comparison results when the mining distances are 94 cm, 112 cm, 130 cm, and 148 cm.

**Figure 11 sensors-20-07215-f011:**
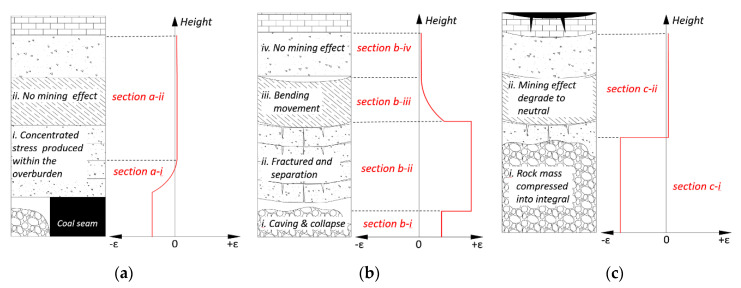
Standardization model of typical overburden deformation characterized by strain distribution: the red line is the strain distribution, and (**a**) shows the strain distribution in the overburden deformation of the leading abutment pressure area, where the mining influence starts developing; (**b**) shows the strain distribution in the overburden deformation of the fracture and separation area where the development of the mining influence is widespread; (**c**) shows the strain distribution in the overburden deformation of the recompacting area where the mining influence tends to be stabilized.

**Figure 12 sensors-20-07215-f012:**
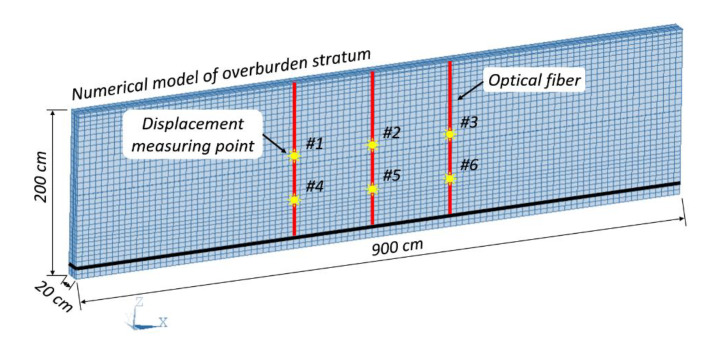
Numerical model of overburden stratum with optical fibers embedding.

**Figure 13 sensors-20-07215-f013:**
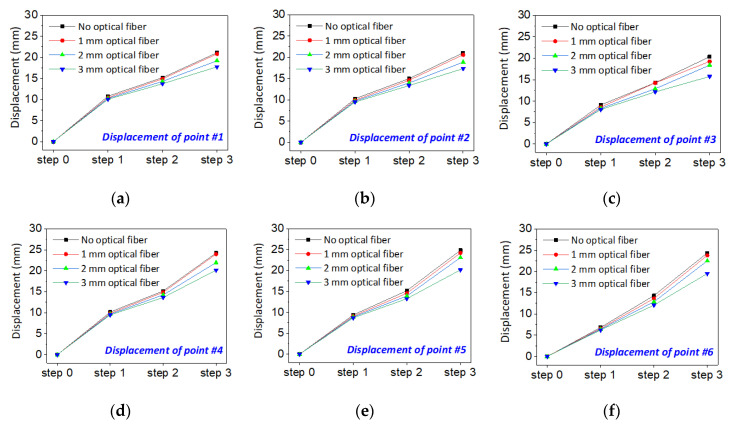
Displacements of overburden stratum in which different diameters of optical fibers were embedded: (**a**–**f**) are results of measuring point #1 to #6.

**Figure 14 sensors-20-07215-f014:**
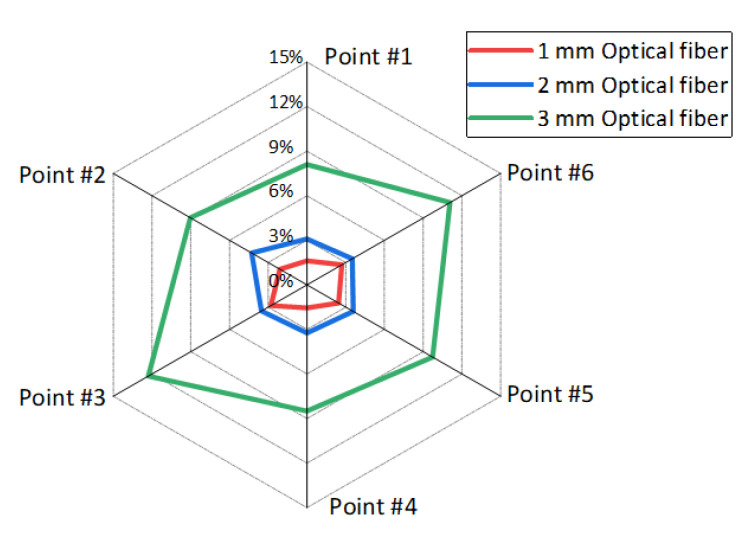
Displacement error rate of overburden stratum in which different diameters of optical fibers were embedded.

**Figure 15 sensors-20-07215-f015:**
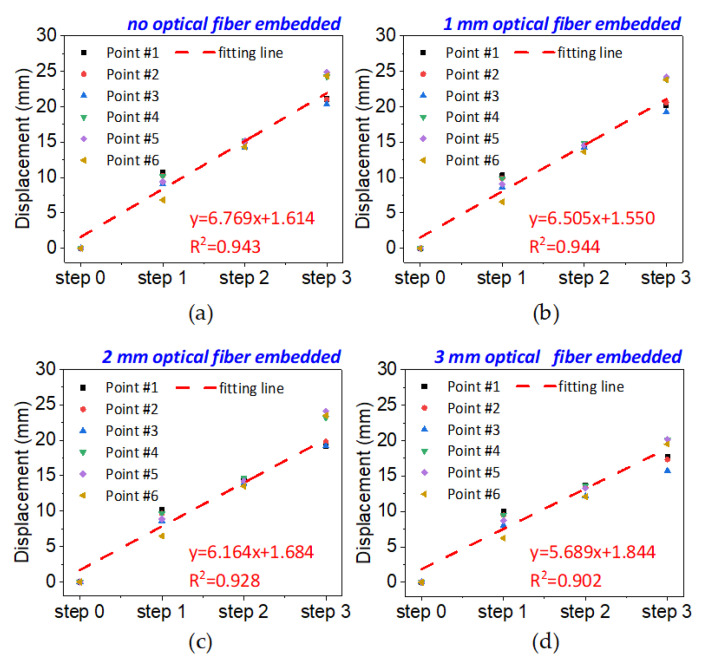
Fitting analysis of mining steps and overburden stratum deformation: (**a**) is the condition of no optical fiber embedded; (**b**–**d**) are the conditions of 1 mm, 2 mm and 3 mm diameters of embedded optical fiber.

**Figure 16 sensors-20-07215-f016:**
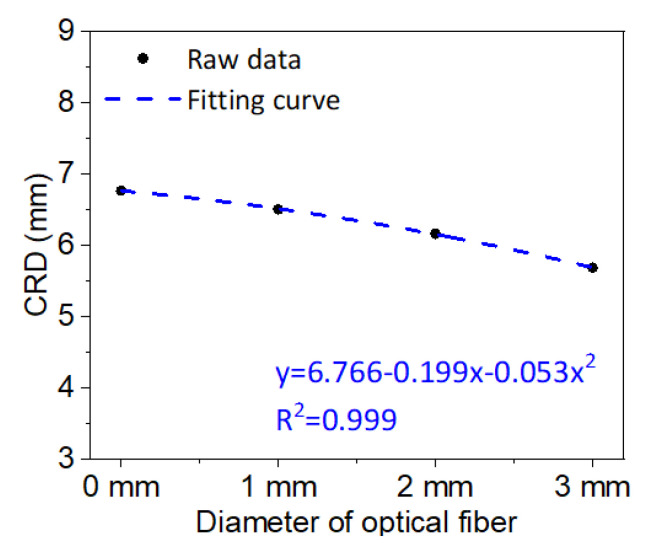
Relationship of optical fiber diameters and the change rate of displacement (CRD).

**Table 1 sensors-20-07215-t001:** Analogous materials of overburden stratum simulation.

Lithology of Simulated Stratum	Simulated Thickness (cm)	Accumulated Height (cm)	Dosage of Analogous Materials (Kg)			Material Ratio
			Sand	Gypsum	Mica Powder	
Mudstone	23	128	198.72	6.62	15.46	937
Kern stone	11	105	73.92	12.67	19.00	746
Conglomerate	8	94	53.76	6.91	16.13	737
Mudstone	17	86	146.88	3.27	13.05	928
Siltstone	37	69	284.16	21.31	49.73	837
Fine sandstone	8	32	53.76	4.61	18.43	728
Siltstone	6	24	46.08	3.46	8.06	837
Kern stone	8	18	30.72	5.27	7.90	746
Coal	5	10				
Mudstone	5	5	43.20	0.96	3.84	928

**Table 2 sensors-20-07215-t002:** Mechanical parameters of the overburden stratum rock mass.

Lithology	Density (Kg/m^3^)	Elastic Modulus (KPa)	Poisson Ratio	Bulk Modulus (KPa)	Shear Modulus (KPa)	Cohesion (KPa)	Friction Angle (°)
Conglomerate	170.23	20.3	0.32	18.81	7.69	25.08	35.00
Mudstone	200.82	48.61	0.36	58.28	17.86	39.07	40.20
Sandstone	170.23	30.16	0.32	27.17	11.47	29.00	38.50
Siltstone	173.29	41.09	0.38	55.68	14.92	34.57	37.60
Coal	96.84	4.47	0.31	3.94	1.70	28.10	23.10

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
