# Peer review of "The Characterization Pattern of Overburden Deformation with Distributed Optical Fiber Sensing: An Analogue Model Test and Extensional Analysis"

_sensors, 2020, doi:10.3390/s20247215_

Round 1
Reviewer 1 Report
This paper innovatively presents that the distributed optical fiber sensing (DOFS) monitoring demonstrates the time-space evolution of overburden deformation and the develop of horizontal three areas and vertical three zones by analyzing measurement results of DOFS and mining overburden deformation states, and constructs the number model to reveal the influence of optical fiber embedding on overburden displacement based on ANSYS simulation analysis. In conclusion, the author’s work is useful but some contents need to be supplement.
- The authors represent the distributed optical fiber sensing (DOFS) acquires the characteristic of overburden deformation, but results of others sensors measurements are not given, the authors had better give some contradistinction to the distributed optical fiber sensing;
- A physical model of overburden stratum by similar material was built in laboratory, which has a length of 300cm, height of 128cm and width of 20cm, and there are 5cm height of floor rock stratum ,5cm height of coal seam and 118cm height of overburden stratum. What is the basis for model parameter setting;
- Since the parameters of the physical model are close to numerical model, the six displacement measure points are marked on the numerical model, what are the actual measure results of these six points by the distributed optical fiber sensing (DOFS), the author had better give some explains;
Author Response
Specific Questions of Reviewer 1 This paper innovatively presents that the distributed optical fiber sensing (DOFS) monitoring demonstrates the time-space evolution of overburden deformation and the develop of horizontal three areas and vertical three zones by analyzing measurement results of DOFS and mining overburden deformation states, and constructs the number model to reveal the influence of optical fiber embedding on overburden displacement based on ANSYS simulation analysis. In conclusion, the author’s work is useful but some contents need to be supplement. 1. The authors represent the distributed optical fiber sensing (DOFS) acquires the characteristic of overburden deformation, but results of others sensors measurements are not given, the authors had better give some contradistinction to the distributed optical fiber sensing; Indeed, contradistinctive analysis is necessary in some study especially in which of measurement demonstration. But in authors’ defense, with all due respect, the comparative analysis had been already carried out in a previous study, which proved that optical fiber sensor has a quite high accuracy. Therefore, an extensional analysis has been proposed in this paper to reveal the deep mechanism of DOFS measurement just on the basis that the consideration of contradistinction had been resolved. And also, as presented in line 152 to line 154 of the original manuscript, it already explained that the measurement of FBG sensor and dial gage was used for accuracy testing compared to optical fiber sensor in the previous study which has been annotated in line 153 as “reference [43]”. In all, the authors thought that it is logical to overlook the comparative of optical fiber sensor and other sensors, because of the contradistinction analysis belongs to the first stage of series studies, and this paper presents the work of the second stage. 2. A physical model of overburden stratum by similar material was built in laboratory, which has a length of 300cm, height of 128cm and width of 20cm, and there are 5cm height of floor rock stratum ,5cm height of coal seam and 118cm height of overburden stratum. What is the basis for model parameter setting; Using a physical model which is constructed with similar materials for regularity study, especially in the field of mining engineering or engineering geology, is a pretty sophisticated approach [1-4]. The parameter settings of the physical model mainly include two aspects, one is the geometrical parameter and the other is the mechanical parameter. But the only thing that matters comes from the simulated object, the geometrical and mechanical parameters have to be set accord with the research object. For instance, in this paper, the ground overburden stratum that affected by underground coal mining is the simulated object, so that in order to simulate the overburden stratum deformation, the geometrical parameters of such model should be set as which has a long length at the lengthwise direction. The height of such model is usually accord with the depth of overburden stratum. The width is commonly changeable based on the laboratory device, if it is much less than the other parameters (i.e. the length and height), it would be a plane model, but if it is consistence with others, it would be a three-dimensional model. In this study, a plane model is suitable to discuss the characterization rules. On the other hand, the mechanical parameters set usually depend on the stress environment, in this paper, the main leading role of stress is the gravity, so that the model materials only have to be set to conform to the overburden stratum based on the similarity law. If the stress environment contains more stress factors besides gravity or the gravity is in weightlessness or overweight, the model test should be carried in a centrifugal machine to conduct a centrifugal model test. More information can be found in the references of the above analysis due to limited space, here only addressed the main factors of the basis of model parameters setting. As a result, the setting of model parameters relies on the simulation requirements of the research object, different objects require different parameters. In this paper, the parameters setting is a universal configuration which is generally used in mining engineering. [1] Li Xiaohong.Simulation technology of rock mechanics[M]. Beijing: Science Press, China,2007. [2] Wang Feng, Xu Jialin, Xie Jianlin. Effects of arch structure in unconsolidated layers on fracture and failure of overlying strata[J]. International Journal of Rock Mechanics and Mining Sciences, 2019, 114, 141-152. [3] Sun Xiaoming, Chen Feng, Miao Chengyu, et al. Physical modeling of deformation failure mechanism of surrounding rocks for the deep-buried tunnel in soft rock strata during the excavation[J]. Tunnelling and Underground Space Technology, 2018, 74, 247-261. [4] Ren Weizhong, Guo Chengmai, Peng Ziqiang, et al. Model experimental research on deformation and subsidence characteristics of ground and wall rock due to mining under thick overlying terrane[J]. International Journal of Rock Mechanics and Mining Sciences, 2010, 47, 614-624. 3. Since the parameters of the physical model are close to numerical model, the six displacement measure points are marked on the numerical model, what are the actual measure results of these six points by the distributed optical fiber sensing (DOFS), the author had better give some explains; Purpose of numerical simulation in this study is to analyze the influence of optical fiber embedding to the analogue model test. Six points arrangement on the numerical model is to contradistinguish the model displacement under the situation of no optical fiber embedding and different diameter optical fiber embedding. As mentioned above (the response of No.1 question), there were already compared the measurement results of DOFS in three points with FBG sensors and dial gages, and it indicated that the DOFS measurement is consistent with FBG and dial gages, so that there are no needs to conduct more comparative studies between DOFS and numerical simulation. Even more, the numerical simulation simplifies the mining periods in only three steps, but the DOFS measurement have more than eighty mining periods, each of them would present different forms under the influence of mining activities. That is also why the DOFS could be a promising method to study the overburden deformation law, cause the DOFS could capture any tiny changes in a wide range. But the numerical simulation is conducted only to compare displacement in only three mining periods, there might be no suitable data could be selected in numerical simulation to compare with the actual data by DOFS measurement. So, in the authors’ opinion, the actual measurement results of these six points by DOFS can be demonstrated. It just needs to capture the data of Figure 3 of the original manuscript at the certain height, then display them by setting the x-coordinate as mining periods (or mining distances) and setting the y-coordinate as strain. There are three optical fibers’ data, each of them can extract two points of data, so that six points of DOFS measurement results would be drawn. More importantly, these results present the same conclusions from another perspective, they have no difference with the illustration of Figure 3, which shows the whole time-space evolution rules in distribution. However, because of the numerical simulation results can not to be used to compare with the actual measurement of DOFS as analyzed above (that is, the numerical simulation only used to compare the influence of optical fiber embedding, the characteristics of overburden deformation might not be simulated precisely, and the numerical model is only close to the analogue model not precisely the same), the consideration of actual measurement results by DOFS in the numerical simulation may be inappropriate, might be even meaningless even more. Nevertheless, in order to respect the reviewer’s comment, the actual measurement of six points by DOFS is presented below in here only (because there cannot be found six points in the analogue model located the same positions compare to the numerical model, here the six points on the analogue model only match to numerical model proportionally). Figure 1. Measurement results of six points of DOFS: the brief summaries of strain distribution at each point are: remains zero at first, increases to a peak, lowering down to negative, remains and maintain stable, which represent the strain change of rock mass at one point under four stages of mining influence (this characterization also can be found in the original manuscript, here it just proves that again in another perspective). Figure 2. FBG measurement results of overburden deformation in previous study: the FBG04, FBG05 and FBG06 are mounted in an analogue model, the height of FBG06 are larger than FBG05, then which of FBG05 are larger than FBG04. The data curve’s peak will go to the right on the x-axis when the FBG sensor mounted higher. As shown in Figure 1, the measurement strain of six points present the law of overburden deformation in the perspective of one-point. The evolution rules of those six results is precisely consistent with the results of FBG sensor measurement as shown in Figure 2 [5] (which is a previous study of the author), cause the FBG sensor only measure change of one point. This analysis further proves that the DOFS measurement is accurate and right for this kind of studies. [5] Chai Jing, Yuan Qiang, Wang Shuai, Li Yi, Zhang Liang. Detection and representation of mining-induced three horizontal zones based on fiber Bragg grating sensing technology[J]. Journal of China University of Mining & Technology, 2015, 44(6): 971-976. So far, the actual measurement results of DOFS at six points have been analyzed in detail.

Reviewer 2 Report
The paper describes an analogue model set up to investigate strain changes induced by excavation in coal mine exploitation. I found the experiment very interesting since it allows to directly observe processes which are hidden on real cases. The manuscript is well organized, even if the English is very poor. A revision from a native speaker is necessary. The title itself is not appropriate and indicative of the paper content. Detailed comments are listed below.
Comments
- In the abstract (Line 18) and overall in the manuscript the Authors refer to their experiment as a “physical modelling experiment”. Indeed, such experiments are in literature known as analogue models. I strongly suggest to the Authors to conform to this concept in order to clearly and directly denote the content of the paper to the readers.
- To enlarge Fig. 2 I suggest to align the two panels vertically .
- (Lines 133-134) “The similar material was comprised by sand, gypsum, mica powder and putty powder with a certain mixed scale.” Please report in a table the materials you are simulating and their analogous material. The materials are shown in Fig.2 but the use of a table would help the reader to better understand.
- Figure captions need to be improved to describe the content of the figures. For example, in caption of Figure 3, I suggest to add that the different line refer to the mining periods.
- The figures (especially Figs. 5,6,7) are too small to see the deformation pattern and the reported indications. Please, enlarge the figure and the fonts.
- (Line 471) “The optical fiber is simulated by the beam 188 unit”. How is the presence of optical fibre simulated? What is it 188 unit? The all sentence is not clear.
- (Line 473) “The constitutive model is the D-P model”. Do you mean Drucker-Prager? Please, clarify.
- The section titles need to be revised as well. They are not clear. For example, for the section “4.3 Influence of optical fiber appllied to overburden deformation model test”, I would suggest “4.3 Influence of optical fiber to the overburden deformation by numerical model”
- Please add the FE model set up: which boundary conditions are applied (displacement or force) along the six boundaries, how the excavation is simulated, which kind of interfaces is used between overburden and fibre (slip, stress free…)
Author Response
The paper describes an analogue model set up to investigate strain changes induced by excavation in coal mine exploitation. I found the experiment very interesting since it allows to directly observe processes which are hidden on real cases. The manuscript is well organized, even if the English is very poor. A revision from a native speaker is necessary. The title itself is not appropriate and indicative of the paper content. Detailed comments are listed below. 1. In the abstract (Line 18) and overall in the manuscript the Authors refer to their experiment as a “physical modelling experiment”. Indeed, such experiments are in literature known as analogue models. I strongly suggest to the Authors to conform to this concept in order to clearly and directly denote the content of the paper to the readers. The reviewer’s suggestions are totally accepted. In order to conform to the literature and academic standard, the “analogue model” and “analogue model test” have been used in the revision paper to replace the “physical modelling experiment”. Meanwhile, in the authors’ defense, the model test study of rock (or soil) deformation in the fields of mining engineering and engineering geology usually be named as model test, similarity simulation or similar material model test, and so on. In the revision manuscript, after full considerations, the author took the reviewer’s suggestion that analogue model test is better demonstration than the physical modelling experiment which could be a potential misleading factor to the readers. 2. To enlarge Fig. 2. I suggest to align the two panels vertically. The Figure 2 has been enlarged and rearranged to show the law of strain distribution clearly and integrally. Also, the figure layout problem of the other figures in revision manuscript have been revised to a better displaying. 3. (Lines 133-134) “The similar material was comprised by sand, gypsum, mica powder and putty powder with a certain mixed scale.” Please report in a table the materials you are simulating and their analogous material. The materials are shown in Fig.2 but the use of a table would help the reader to better understand. A table of analogous material parameters (the Table 1 below) has been supplemented into the revision manuscript to show the analogous materials and its ratios. Table 1. Analogous materials of overburden stratum simulation. Lithology of simulated stratum simulated thickness (cm) Accumulated height (cm) Dosage of analogous materials (Kg) Material ratio Sand Gypsum Mica powder Mudstone 30 128 259.20 8.64 20.16 937 Sandstone 14 98 94.08 16.13 24.19 746 Mudstone 8 84 53.76 6.91 16.13 737 Sandstone 14 76 94.08 4.03 36.29 719 Mudstone 9 62 77.76 1.73 6.91 928 Siltstone 10 53 76.80 5.76 13.44 837 Conglomerate 6 43 51.84 1.73 4.03 937 Mudstone 6 37 51.84 1.15 4.61 928 Sandstone 8 31 53.76 4.61 18.43 728 Conglomerate 6 23 40.32 5.18 12.10 737 Siltstone 7 17 53.76 4.03 9.41 837 Coal 5 10 Sandstone 5 5 43.20 1.44 3.36 937 4. Figure captions need to be improved to describe the content of the figures. For example, in caption of Figure 3, I suggest to add that the different line refer to the mining periods. The caption of Figure 3 has been revised according to reviewer’s suggestions. Besides, the all other captions also have been checked and revised to ensure they describe the figure’s content appropriately and accurately. 5. The figures (especially Figs. 5,6,7) are too small to see the deformation pattern and the reported indications. Please, enlarge the figure and the fonts. The Figure 5, 6 and 7 have been enlarged and rearranged. And all the other figures in revision manuscript have been checked and modified to a proper size that can be clearly seen its illustrations and indications. 6. (Line 471) “The optical fiber is simulated by the beam 188 unit”. How is the presence of optical fibre simulated? What is it 188 unit? The all sentence is not clear. The beam 188 unit is a beam unit based on the Timoshenko beam theory in the ANSYS software, it had been built in the overburden stratum model to simulate the optical fiber cause its geometrical and mechanical properties is close to the optical fiber. However, the description of numerical simulation of overburden stratum and optical fiber has been rewritten in the revision manuscript to expressly and fully introduce the numerical simulation. 7. (Line 473) “The constitutive model is the D-P model”. Do you mean Drucker-Prager? Please, clarify. Indeed, the Drucker-Prager model is commonly used in rock and soil mechanics so that it has been simplified to D-P model, which is not a formal expression and easily induce misunderstanding. The Drucker-Prager has been widely used in rock mechanics and geotechnical engineering, in this study, the overburden stratum rock mass material has been set to submit the Drucker-Prager yield criterion, which defines the stress-strain relationship of the numerical model material. It has been clarified in the revision paper and recovered to its full description. 8. The section titles need to be revised as well. They are not clear. For example, for the section “4.3 Influence of optical fiber applied to overburden deformation model test”, I would suggest “4.3 Influence of optical fiber to the overburden deformation by numerical model”. The title and the all subtitles of the revision manuscript have been checked and revised to present a clear (better understanding) description of each contents. 9. Please add the FE model set up: which boundary conditions are applied (displacement or force) along the six boundaries, how the excavation is simulated, which kind of interfaces is used between overburden and fibre (slip, stress free…) A new paragraph of the numerical simulation setup has been supplemented into the revision manuscript (which can be found as follow in this document), which presents the detailed information about the modelling, boundary conditions, interface setting and excavation. “The optical fiber was simulated by the BEAM 188 unit of ANSYS software, which is a three-dimensional linear unit based on the Timoshenko beam theory and usually be used for simulating the slender beam due to it considers the shear deformation and have six degree of freedom in each node. In this case, given that the optical fiber has a much larger length than the cross section, the BEAM 188 is just suitable for simulating. The Drucker-Prager model has been used for defining the overburden stratum material because of it is fit for simulating the elastoplastic characteristics of rock and soil. Then, free boundary had been applied to the top side of the model and the other five sides were applied with displacement constraint. Thereafter, the optical fiber was bonded with the overburden stratum and the kill element has been applied to simulate the coal seam excavation.”
